# New Insights on the Conversion Reaction Mechanism in Metal Oxide Electrodes for Sodium-Ion Batteries

**DOI:** 10.3390/nano11040966

**Published:** 2021-04-09

**Authors:** Jadra Mosa, Francisco José García-García, Agustín R. González-Elipe, Mario Aparicio

**Affiliations:** 1Instituto de Cerámica y Vidrio (CSIC), Kelsen 5, 28049 Madrid, Spain; jmosa@icv.csic.es; 2Dpto. Ingeniería y Ciencia de los Materiales y del Transporte, Escuela Técnica Superior de Ingeniería, Universidad de Sevilla, Camino de los Descubrimientos, 41092 Sevilla, Spain; fgarcia49@us.es; 3Instituto de Ciencia de Materiales de Sevilla (CSIC–Universidad de Sevilla), Avda. Américo Vespucio 49, 41092 Sevilla, Spain; arge@icmse.csic.es

**Keywords:** Na-ion batteries, anode, conversion reaction, magnetron sputtering, nickel oxide

## Abstract

Due to the abundance and low cost of exchanged metal, sodium-ion batteries have attracted increasing research attention for the massive energy storage associated with renewable energy sources. Nickel oxide (NiO) thin films have been prepared by magnetron sputtering (MS) deposition under an oblique angle configuration (OAD) and used as electrodes for Na-ion batteries. A systematic chemical, structural and electrochemical analysis of this electrode has been carried out. The electrochemical characterization by galvanostatic charge–discharge cycling and cyclic voltammetry has revealed a certain loss of performance after the initial cycling of the battery. The conversion reaction of NiO with sodium ions during the discharge process to generate sodium oxide and Ni metal has been confirmed by X-ray photoelectron spectra (XPS) and micro-Raman analysis. Likewise, it has been determined that the charging process is not totally reversible, causing a reduction in battery capacity.

## 1. Introduction

Lithium-ion batteries have been rapidly developed and currently dominate the secondary battery market. However, their use is reaching its limit due to the shortage and high price of lithium compounds. Because of the massive energy storage associated with renewable energy sources, sodium-ion batteries have the advantage of using an abundant and low-cost exchanged metal [1,2,3,4]. In this context, although carbon materials are the most widely used components of anodes in lithium-ion batteries, different studies indicate a limited insertion capacity of sodium ions in these types of materials [5,6,7]. Numerous alternative anodes have been studied for years in sodium-ion batteries. One of the most common approaches entails the use of transition metal oxides. Nickel oxide (NiO) is active against sodium ions, but unlike other oxides, the metal (Ni) produced during its reduction does not form an alloy with sodium [1]. In this case, the charge–discharge process is carried out only through the following single-stage conversion reaction:NiO + 2 Na <==> Na_2_O + Ni(1)

This process has been demonstrated experimentally by the in situ Transmission Electron Microscopy (TEM) observation during the charge–discharge cycling of the formation of Ni nanograins within a Na_2_O matrix [8].

Multielectron exchanges in this type of conversion reaction provide high-energy densities but also an insufficient cycling stability and large hysteresis due to their low electrical conductivity and significant volume changes during cycling. The development of materials with an adequate pore distribution and interconnectivity is essential to overcome these constraints [9,10]. Under a similar scheme, it has been proven that the reduction in NiO’s particle size when using this material as an anode in Li-ion batteries contributes to an increase in the energy efficiency [11,12,13], an improvement that is based on the reduction in the hysteresis loop between the charge and discharge stages.

Different articles in the literature have examined the possibility of combining NiO with other oxides [14,15] in order to achieve a synergistic effect and improve its electrochemical performance as an electrode for Na-ion batteries. In addition, it has been shown that the low electrical conductivity of oxides usually makes it necessary to incorporate conductive materials such as carbon or noble metals in the fabrication of electrodes. Numerous examples exist in the literature about this approach combining various oxides, most commonly NiO, with carbon materials, both in nanometric and highly porous forms [9,16,17,18,19,20,21,22]. High-current density values and an improved reversibility behavior were found for the tested combinations.

In the field of metal oxide electrode layers, M. C. López et al. [23] evaluated a NiO thin film prepared by electrodeposition and thermal oxidation as a potential anode for Na-ion batteries. Although the first discharge showed a high-capacity value (around 800 mA h g^−1^), the capacity retention was poor with substantial drops in this parameter after eight cycles. Another example consisted of thin films of transition metal oxides (NiO and Fe_2_O_3_) prepared by electrodeposition [24], whose galvanostatic cycling in sodium half cells resulted in a discharge capacity of around 500 mA h g^−1^ during the first cycle. From the knowledge gained in this work, it appears that improving the electrochemical behavior of the sodium batteries requires an efficient management of the oxide thin film’s porosity, a possibility that can be done straightforwardly with the magnetron sputtering (MS) technique under an oblique angle deposition configuration (MS-OAD) [25,26].

This work is primarily motivated by the aim of developing simple and reliable procedures for the synthesis of metal anodes suitable for sodium-ion microbatteries. In this context, we propose the synthesis and analysis of the behavior of a new kind of porous and nanocolumnar NiO electrode without the presence of other materials such as carbon that may also intervene in the process of the insertion–extraction of sodium ions. This analysis should provide new insights about the mechanism of interaction of sodium ions with porous NiO electrodes and the processes that usually lead to the commonly observed drop in capacity with an increase in the number of cycles. For this purpose, a porous and nanocolumnar NiO thin film has been deposited by MS-OAD. MS is particularly favored in mass production and is easy to implement both in industry and for basic research. MS-OAD is achieved by placing the substrate at a glancing angle with respect to the target, a configuration that renders thin films with a columnar and highly porous microstructure with mesopores extending from the surface up to the interface with the substrate [27]. This method has been successfully applied to obtain porous nanostructured thin films [28] with enhanced performances as electrocatalysts [29], electrochromic electrodes [30], nanosensors [31], solid oxide fuel cells [32], and Na-ion battery electrodes [33], the latter consisting of WO_3_ anodes that showed a great robustness and reproducibility. The present work contains a description of the synthesis procedure of NiO electrodes by MS-OAD, a physicochemical characterization of the film electrode (as-prepared and after being subjected to various electrochemical cycles), and a thorough analysis of their electrochemical performance as electrodes in Na-ion batteries.

## 2. Materials and Methods

NiO thin films were prepared by reactive pulsed Direct Current DC MS-OAD using a nickel target of 50 mm diameter and 0.3 mm thickness. The magnetron head was operated at a power of 200 W and a pulsed voltage of 550–600 V at a frequency of 80 kHz. The base pressure in the chamber prior to the deposition and during the deposition was 3 × 10^−6^ and 5 × 10^−3^ mbar, respectively. The plasma gas consisted of an O_2_/Ar mixture at a mass flow ratio of 0.06. The angle between the target, in a horizontal and upper position, and the perpendicular substrate was 80° and the distance between the center of the target and the substrates 10 cm. Silicon, soda lime glass, and high mirror-polished copper were used as substrates to carry out different characterization studies as indicated in the text.

The microstructure of the films deposited on doped silicon (100) wafers was examined by scanning electron microscopy (SEM) using a Hitachi S4800 field emission microscope. The samples were diced for cross-sectional analysis. Rutherford backscattering (RBS) spectra were recorded in a tandem accelerator (Centro Nacional de Aceleradores, Universidad de Sevilla, Spain) with α particles of 2.0 MeV, a passivized implanted planar silicon (PIPS) detector located at a 165° scattering angle, with a beam current and diameter of 1.7 nA and ~1 mm, respectively. The spectra were analyzed with the SIMRA6.0 program. X-ray photoelectron spectra (XPS), taken to chemically characterize the surface of the samples, were recorded in a PHOIBOS spectrometer working in the constant-pass energy mode at a value of 20 eV. The binding energy (BE) scale of the spectra was referenced to the C1s peak of the spurious carbon contaminating the surface of the samples at a value of 284.5 eV. X-ray diffraction (XRD) patterns were obtained using a Panalytical X’Pert Pro diffractometer with Bragg–Brentano geometry and an X’Celerator detector. The diffraction patterns were recorded from 30° to 120° (2θ) with a step size of 0.05° and a counting time of 300 s per step. The analysis was performed using the X’Pert HighScore Plus software.

The characterization of the NiO samples by XPS was carried out both for their “as-prepared” state and after their usage as electrodes. In this case, two samples were selected for analysis. A NiO-3V sample was taken from the electrochemical cell after cycling the voltage several times with the voltage set at 3V. Under these conditions, it is assumed that the electrode was “charged” and the Equation (1) set to the left. A NiO-0V sample was taken out from the cell after the same number of cycles but in a “discharged” state as expected by the fact that the applied voltage at that moment was 0.01 V (Equation (1) was displaced to the right). These samples were carefully handled after removing them from the electrochemical cell inside the glove box: they were soaked in propylene carbonate; carefully dried; and stored in a desiccator until they were introduced to the XPS analysis chamber. These electrodes were analyzed before and after a mild sputtering treatment with Ar^+^ ions of 1000 eV kinetic energy. The utilized ion dose, as calibrated with a standard, was equivalent to that required to remove a layer of approximately 5 nm thickness in the case of homogeneous and compact samples.

The Raman analysis was carried out using a confocal micro-Raman (Witec alpha-300R) with a laser excitation of 532 nm and a 100× objective lens (NA = 0.9) for coatings deposited on Cu substrates. The optical resolution of the confocal microscope is limited to 200 nm laterally and 500 nm vertically, while the Raman spectral resolution is limited to 0.02 cm^−1^. The spectra were acquired using the as-prepared samples and after their electrochemical characterization (at the full-charge and discharge electrodes).

An electrochemical characterization of the NiO films prepared on Cu substrates was carried out in an argon-filled glass beaker cell using Na foils as counter and reference electrodes and 1M NaClO_4_ in propylene carbonate (PC) as the electrolyte. Galvanostatic charge-discharge cycles (Multichannel Potentiostat VMP3, Biologic) were achieved between 0.010 and 3.000 V vs. Na^+^/Na at different current densities. A cyclic voltammetry was performed using the same voltage range at 5.0 mV min^−1^.

## 3. Results

### 3.1. Electrode Thin Film Characterization

SEM micrographs of the NiO MS-OAD thin film deposited on silicon for both the cross-section and planar views are shown in Figure 1a,b, respectively. This NiO film exhibits a columnar microstructure with nanocolumns tilted at 71° with respect to the normal surface. The nanocolumns are about 560 nm in length and 75 nm in diameter and are separated by large voids and pores. It is worth noting that the presence of the tilted orientation of these nanocolumns is a characteristic feature of MS-OAD thin films, as a result of the enhancement of shadowing effects at the nanoscale during the deposition process [27].

The RBS experimental and simulated curves of the NiO MS-OAD are displayed in Figure 2a. RBS confirms the homogeneous distribution of nickel and oxygen atoms throughout the whole film’s thickness and the completely oxidized state of nickel, i.e., NiO. This is also confirmed by the Ni2p photoelectron spectrum of the as-prepared sample (Figure 2b), depicting a shape characteristic of this compound [34].

The density of the NiO MS-OAD thin film can be estimated using the mass thickness determined by RBS (i.e., the number of atoms per cm^2^) and the thickness can be determined from the SEM cross-section image (Figure 1a). The calculated film density, 3.67 g cm^−3^, yields an estimated porosity of ca. 45% of the total volume. The XRD pattern of the NiO MS-OAD film is depicted in Figure 2c. The main diffraction peaks appear at 37.1°, 43.2°, 62.8°, 74.9°, and 79.2° and are assigned to NiO (JCPDS card no. 44-1049). Figure 2d shows the Raman spectrum of the as-prepared NiO thin films. It depicts NiO vibration modes at around 200 cm^−1^, 450 cm^−1^, 571 cm^−1^, 787 cm^−1^, 1093 cm^−1^, and 1457 cm^−1^ that can be attributed to one phonon (1P), one-phonon transverse optical (TO), one phonon and one magnon (1P + 1 M), two-phonon transverse optical (2TO), 2LO, and two-magnon (2 M) modes, respectively [35,36,37,38,39,40]. The detection of these six bands confirms the existence of a NiO phase, while the low intensity of the TO mode and the absence of a characteristic TO + LO mode indicates the nano-crystalline nature of the samples [38].

### 3.2. Electrochemical Performance

The first cathodic scan of the cyclic voltammetry (CV) curve (Figure 3) shows two peaks located at 0.55 and 0.30 V. These peaks can be associated with the conversion reaction of NiO to Ni^0^, the precipitation of Na_2_O, and the formation of a solid electrolyte interface (SEI) at the surface of the NiO electrode [10,16,17,18,19,20,21,22]. These cathodic peaks decrease in intensity with the cycle’s number, indicating the end of the SEI formation and the stabilization of the Na^+^ insertion-extraction processes. A similar behavior was observed [10], associating the irreversible sodium-ion consumption to side reactions of the formation of the SEI layer. These assignments of the different peaks of the cyclic voltammetry also correlate to those of ultrathin NiO nanosheets (4–5 nm in thickness) synthesized via a solvothermal process followed by annealing in air [16]. Meanwhile, the anodic scans show a very wide band at around 1.4 V, assigned to the conversion of Ni^0^ to NiO and Na_2_O to Na^0^, and a feature close to 3.0 V during the first two cycles, possibly associated with the interaction of the liquid electrolyte with the sodium anode based on the relative stability of the carbonate solvents at high voltages. After the first cycle, the CV curves are very similar, indicating an adequate electrochemical reversibility with these measurement conditions.

Figure 4a shows the galvanostatic discharge–charge curves (capacities vs. voltage) at 400 mA g^−1^ between 0.010 and 3.000 V (vs. Na^+^/Na). The first discharge scan reaches a huge capacity value of 6305 mA h g^−1^, associated mainly with the SEI formation. The electrochemical evaluation of the NiO nanosheet’s anode using galvanostatic charge–discharge profiles also shows a very high discharge capacity during the first cycle, which is associated with the SEI formation and electrolyte decomposition [16]. Increasing the cycle’s number leads to a continuous reduction in the discharge capacity due to the completion of the SEI formation, but also because some amount of Ni^0^ and Na_2_O produced during the discharge process remains in the electrode and does not participate in the charge process and subsequent discharges (see next section). A similar analysis using NiO nanosheets shows a decreasing discharge capacity due to the formation of irreversible Na_2_O, more SEI film formation, and a serious electrical isolation caused by the large size of Na^+^ in comparison with Li^+^ [17]. Also, in this paper, the large capacity difference between the initial cycles is attributed to the decomposition of the electrolyte and the irreversible formation of the SEI film, which aligns with the CV results. The plots of charge and discharge capacities vs. cycle number (Figure 4b) clearly show this behavior. The reduction in the discharge capacity with the number of cycles in comparison to the rather stable value shown by the charge capacity, together with the higher discharge capacity values, seem to confirm that not all the Ni^0^ formed during the discharge processes participates in the charge steps. This situation originates a lower amount of available NiO after each successive discharge process, leading to a stabilized final discharge capacity of 467 mA h g^−1^ after 16 cycles.

The galvanostatic discharge–charge curves (capacities vs. voltage) at 1000 mA g^−1^ between 0.010 and 3.000 V (vs. Na^+^/Na) are presented in Figure 5a. This sample was previously charged–discharged three times at 400 mA g^−1^ in order to consolidate the SEI before evaluating the cycling behavior at 1000 mA g^−1^.

The curves show a performance similar to that observed during the cycling at 400 mA g^−1^, with a decrease in the capacity. The explanation relies again on the incomplete reversibility of the conversion between NiO and Ni^0^. In this case, as a result of the higher discharge and charge rates, the value of the discharge capacity after 40 cycles was 169 mA h g^−1^. This same sample, after the 40 charge–discharge cycles at 1000 mA g^−1^, was evaluated at 1200 mA g^−1^ for 50 cycles and then at 800 mA g^−1^ for 50 cycles (Figure 5b) in order to analyze the stability of the electrode. The observed continuous reduction in the discharge and charge capacity values is attributed to the certain partial irreversibility of the Na^+^ ion during the insertion–extraction process.

Long-term cycling at 1000 mA g^−1^ was also performed in order to analyze in detail the stability of the NiO electrodes (Figure 6). The figure shows the trend already observed in the previous figures with a rapid reduction in the capacity with the number of cycles. This decrease is particularly well-observed up to approximately 300 cycles and is possibly due to the aforementioned continued reduction in the amount of available NiO. In the last 150 cycles, the capacity values remain almost constant, which could be associated with a stabilization of the redox reactions at the electrode surface.

### 3.3. Post-Mortem Analysis of the Electrodes

To further test the hypothesis of a reduction in the NiO to metallic Ni while sodium becomes incorporated in the electrode during cycling (c.f., Equation (1)), we have investigated by XPS the state of the electrodes at different stages of the cycling process. The general spectra of the NiO-3V and NiO-0V samples, as taken from the electrochemical cell, revealed the presence of a high concentration of Na, Ni, and C, together with oxygen. The first analysis of the characteristic zone spectra of these elements confirmed that sodium (i.e., Na (1s) band) was in a cationic form and carbon (C1s peak) had a complex shape with two main components attributable, respectively, to adventitious carbon (BE. 284.5 eV), due to contamination by exposure to air before the incorporation of the samples in the spectrometer chamber, and a carbonate-like species (road band around 289.3 eV), likely due to the carbonation of the sodium oxide/hydroxide incorporated in the electrodes according to (1) or from the electrolyte. Meanwhile, as reported in Figure 2b, the Ni2p spectra in these samples presented a complex shape, resulting from the contribution of Ni^0^ and Ni^2+^ species and characterized by the Ni2p3/2 main contributions at approximately 852.5 and 856.0 eV, respectively [39]. Due to the exposure of the electrodes to air before their analysis by XPS, the detected partial oxidation of nickel at the surface of these samples might be due to air oxidation. To avoid this problem and to reduce surface contamination effects, we subjected to samples to a mild Ar^+^ sputtering treatment (see experimental section for details) that, under the working conditions of the calibrated ion gun, is equivalent to the removal of the outmost 5 nm surface layer. We assume that the spectra after this mild sputtering are a better representation of the chemical state and composition of the bulk of the electrodes. Figure 7a shows that in the NiO-0V sample, the shape of the Ni2p spectrum after sputtering corresponds to that of Ni^0^, while in the NiO-3V sample, the spectrum (Figure 7b) indicates the presence of both Ni^0^ and NiO [41]. Interestingly, the Ni/Na atomic ratio determined from the analysis of the spectra presented values of 0.35 and 3.35 for the NiO-0V and NiO-3V samples, respectively. These values support a considerable incorporation of sodium in the bulk of the electrodes in the NiO-0V sample where, in agreement with Equation (1), nickel appears in a metallic form. In the NiO-3V sample, the amount of incorporated sodium is much smaller, very likely corresponding to a residual amount of sodium ions that remains in the electrode, likely causing the progressive decrease in performance indicated in Figure 4, Figure 5 and Figure 6. In agreement with the residual character of the sodium remaining in this electrode, the chemical state is a mixture of Ni^0^ and NiO (note that the relative amount of Ni^0^ is likely enhanced due to the partial reduction of NiO to Ni^0^, known to occur when NiO is subjected to Ar^+^ bombardment [42]).

In addition, we used confocal micro-Raman (CRM) coupled with Atomic Force Microscope (AFM) and integrated software tools of a NiO electrode after charge (NiO-3V) and discharge (NiO-0V) processes to investigate the redox mechanism. The Raman spectra of NiO thin films after charge and discharge are shown in Figure 8a.

The spectra of the charged NiO thin film are quite similar to those of the as-prepared NiO (Figure 2d), since the Na ions leave the electrode while Ni is oxidizing. However, the discharged NiO thin film presents significant differences that justified a more in-depth study of this sample. The Raman image of Figure 8b is an XY surface mapping of the discharged NiO electrode (NiO-0V) where two different spectra can be observed and combined in a color-coded image. Both color zones, blue and red, correspond to the curves of the same color in Figure 8c. The blue zone and spectra clearly reveal a minority region showing four peaks that can be discerned in the spectra at 466, 621, 820, and 1093 cm^−1^ [35,37]. Two vibration modes, the small and narrow peak at 200 cm^−1^, related to a one-phonon excitation, and the band, related to a two-magnon excitation at 1457 cm^−1^, have disappeared in comparison with the results for the as-prepared NiO electrode. The red spectrum shows two vibration bands at 621 and 1093 cm^−1^ related to Ni, and an intense Raman peak at 278 cm^−1^ that can be indexed to the Ni–Na mode [40,41]. This result suggests the formation of Ni and Na_2_O during the discharge process as previously discussed.

## 4. Discussion

NiO thin films (560 nm thick) have been successfully prepared by magnetron sputtering (MS) deposition under an oblique angle configuration (OAD). NiO films show a columnar microstructure of about 75 nm in diameter, separated by large voids and pores. This architecture facilitates the diffusion of sodium ions as the electrolytes can penetrate into the empty spaces, and it accommodates the volume changes caused by the successive charge and discharge cycles. XPS and a micro-Raman analysis of the charged and discharged samples have confirmed the reversibility of the conversion reaction of NiO with sodium ions. The results of the electrochemical study indicate the high discharge capacities of NiO films during the first cycle at different current intensities. However, the continuous reduction in capacity with the number of cycles is a consequence of a limited reversibility in the charge-discharge cycling, as demonstrated by XPS and the micro-Raman analysis. The accessibility to NiO particles inside the nanocolumns with a well-defined porous structure could improve the electrochemical behavior of these electrodes. On the other hand, an enhancement of the electrical conductivity (not considered in this paper) should also lead to higher capacity values.

## Figures and Tables

**Figure 1 nanomaterials-11-00966-f001:**
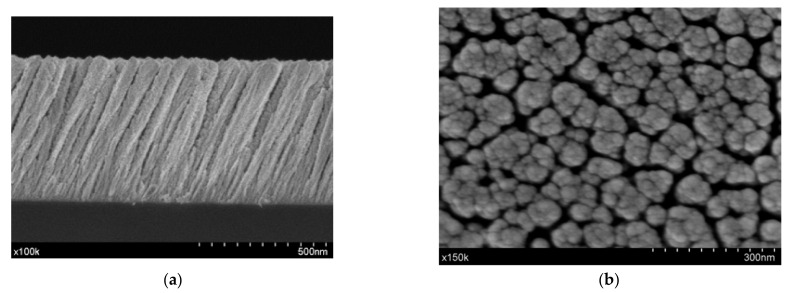
Scanning electron microscopy (SEM) microstructure images of nickel oxide (NiO) magnetron sputtering technique under an oblique angle deposition configuration (MS-OAD): cross-section (**a**) and top view (**b**).

**Figure 2 nanomaterials-11-00966-f002:**
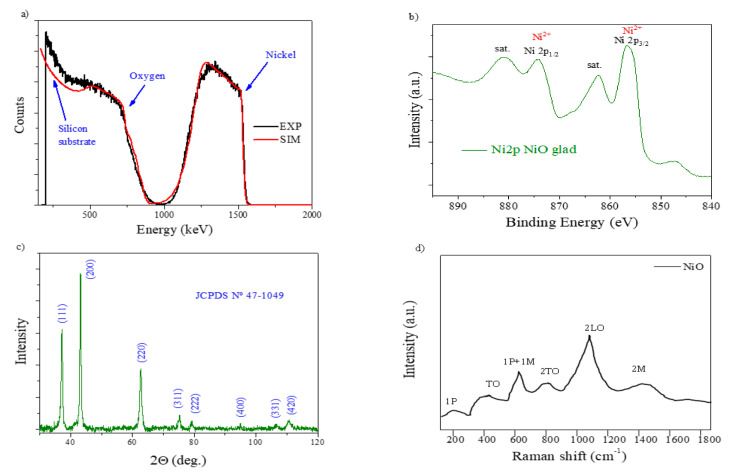
The physicochemical characterization of the as-prepared NiO thin film: (**a**) Rutherford backscattering (RBS); (**b**) X-ray photoelectron spectra; (**c**) X-ray diffraction (XRD); and (**d**) confocal micro-Raman.

**Figure 3 nanomaterials-11-00966-f003:**
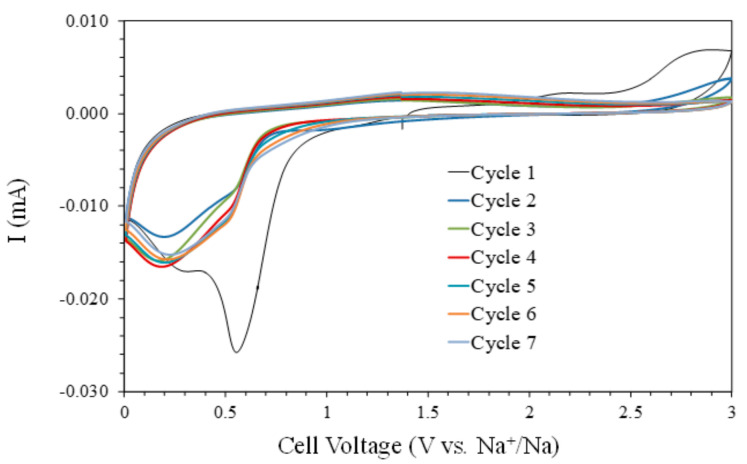
The cyclic voltammetry curves (5 mV min^−1^) of the NiO films prepared on Cu substrates.

**Figure 4 nanomaterials-11-00966-f004:**
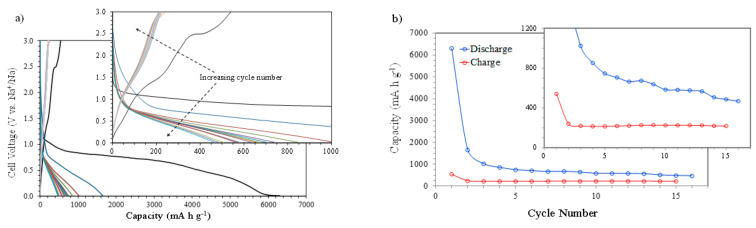
The electrochemical characterization of NiO films at a current intensity of 400 mA g^−1^: (**a**) discharge–charge profiles (capacity vs. cell voltage) and (**b**) discharge and charge capacities vs. cycle number.

**Figure 5 nanomaterials-11-00966-f005:**
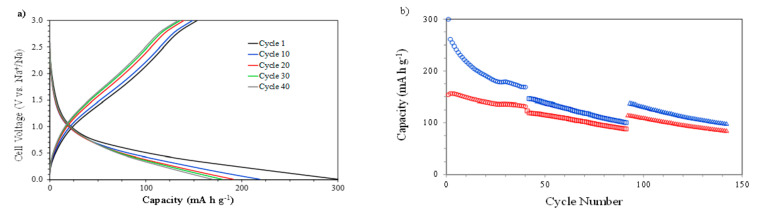
The electrochemical characterization of NiO films at different current intensities: (**a**) discharge–charge profiles (capacity vs. cell voltage) at 1000 mA g^−1^ and (**b**) discharge (blue) and charge (red) capacities vs. cycle number at 1000 (o), 1200(□), and 800 (Δ) mA g^−1^.

**Figure 6 nanomaterials-11-00966-f006:**
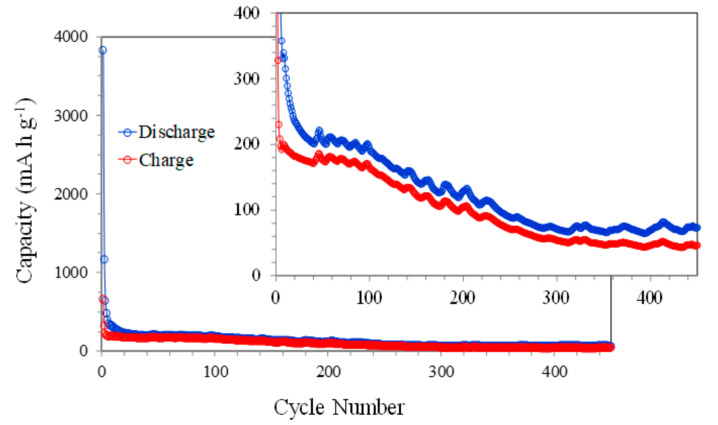
The discharge-charge capacities vs. cycle number of NiO films at 1000 mA g^−1^.

**Figure 7 nanomaterials-11-00966-f007:**
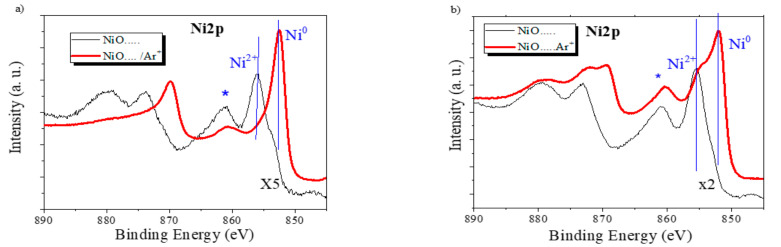
The Ni2p photoelectron spectra of the NiO-0V sample (**a**) and the NiO-3V sample (**b**) before (thin black line) and after (red thick line) a mild sputtering with Ar+. (Note that the spectra before sputtering have been multiplied by the factors indicated in the plot). The Ni2p3/2 features associated with Ni^0^ and Ni^2+^ and a satellite that is particularly intense for the Ni^2+^ species (marked with an asterisk) are highlighted in the plots.

**Figure 8 nanomaterials-11-00966-f008:**
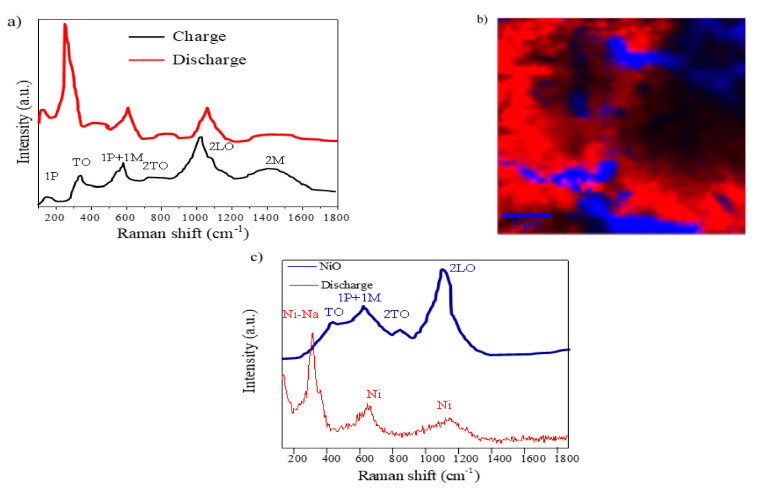
A confocal micro-Raman analysis: (**a**) the Raman spectra of NiO-0V and NiO-3V films; (**b**) the Raman image of a discharged NiO film (integration time of 0.03 s); the colors in the spectra correspond to different areas in the Raman image using a filter for 278 cm^−1^; and (**c**) the main Raman spectra associated with the two different colors.

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
