# Peer review of "New Insights on the Conversion Reaction Mechanism in Metal Oxide Electrodes for Sodium-Ion Batteries"

_nanomaterials, 2021, doi:10.3390/nano11040966_

Round 1

Reviewer 1 Report

The subject is of interest for the research community working in this area and the experimental work is adequately described. However, this is not the first paper on this topic. As a consequence, a more detail comparison with available literature data is required and the authors are suggested to emphasise the difference of their work in comparison to published data.

The Authors state that the continuous reduction in capacity with the number of cycles is a consequence of a limited reversibility in the charge-discharge cycling. Have they totally excluded the possibility of a partial influence of the morphology (i.e. collapse in the nanostructure)? Post-mortem SEM analysis to confirm that the nanostructure is maintained after cycling is necessary.

Author Response

A more exhaustive comparison has been made with results in the literature, especially with those articles that are based on NiO fims, although prepared by other processing methods. The purpose has been to confirm the processes that take place during the charge-discharge cycling. Consequently, new text has been added in the Results section.

We agree with the reviewer that a possible change in NiO layer morphology may have occurred during charge-discharge cycling. This change could be one of the causes of the limited reversibility observed during the tests. Likewise, we agree that a post-mortem SEM study may be useful to study this point. We intend to carry out this study in the immediate future, but the time provided to carry out the review of the manuscript (7 days) makes it impossible to do it for this article.

Reviewer 2 Report

This work is devoted to the synthesis of NiO thin film by a magnetron sputtering deposition method under an oblique angle configuration and disclosing NiO's conversion reaction mechanism with sodium ions. Therefore, I recommend accepting this article after revision as below:

(1) For the sentence in the introduction “NiO is part of a group of oxides in which sodium is inactive, that is, it does not form an alloy with nickel metal”, I believe the authors want to express that Ni rather than NiO is inactive in sodium storage. So, the sentence should be corrected.

(2) As mentioned in the manuscript, the anodic peak around 3.0 V is associated with a partial decomposition of the SEI layer that stabilizes in subsequent cycles. How could authors demonstrate that the anodic peak is associated with a partial oxidative decomposition of the SEI layer? And why would it be stable in subsequent cycles?

(3) As is well known, the products/intermediates and SEI layer formed during the discharge/charge process are sensitive to air. In this manuscript, the electrodes after cycling have been exposed to the air before doing the XPS measurements. A transfer setup can be employed to avoid the contact between electrodes and air. Some related data should be retested to make the manuscript more scientific.

(4) In/Ex-situ XRD test is an effective method to study the phase transition process. So, some related measurements should be supplemented to demonstrate the conversion reaction mechanism further.

Author Response

1) In this sentence, our intention has been to indicate that the Ni0 formed during the discharge process does not form an alloy with Na as it does when using other oxides. The sentence has been modified.

2) We agree that the explanation is not clear at all.We believe that the peak during the oxidation process at high voltages in the first two cycles is due to the interaction of the electrolyte with the metallic sodium based on the limited stability of carbonate-based solvents under these conditions.Modifications have been made to the original text to try to explain it adequately.

3) The reviewer is right and we should have used a device to prevent exposure of the NiO electrode to air prior to the XPS measurements.For this reason, measurements were made directly on the surface of the electrode and also after a previous sputtering treatment to avoid the surface area affected by its interaction with air.In the new tests that we carry out, we will use a device to avoid this exposure to air.

4) Again we agree with the reviewer, and additional characterization techniques to those used would be very useful to complement the results.This comment, as well as the previous ones, will be taken into account in future works.It is not possible to do them at this time due to the short time available to review the article (seven days).

Reviewer 3 Report

The suggested manuscript describes an efficient magnetron-sputtering method for the preparation of nanocolumnar NiO film with controllable porosity and morphology for possible use as an anode for sodium-ion batteries.

The preparation method and the evaluation/characterization techniques are described in detail and allow their reproduction. On the other hand, the nano-columnar morphology and the appropriate porosity (45 %) in fact did not improve significantly the electrochemical performance of this anode. The reason is that only these parameters are not and cannot allow sufficient reversibility of the electrochemical reaction NiO + 2Na <==> Ni + Na2O. The reason for the irreversibility of this reaction is intrinsic: gradual growth of isolated “islands” of Ni, which quickly become inactive. This feature would not change regardless morphology, porosity, and other similar factors of purely physical origin of the NiO material. Nevertheless, the study could be of interest for the readership of this Journal and I would recommend it for publication provided the authors delete their last statement in the conclusions:

“The accessibility to NiO particles inside the nanocolumns, more empty space available for conversion between oxides and elements, and electrical conductivity should be optimized in order to increase the capacity stability.”

And replace it with:

“Despite the appropriate and well-controlled morphology the cycle life and the coulomb efficiency of the NiO nano-columnar anodes did not improve significantly. Reversibility of the electrochemical conversion reaction between NiO and Na+ is unlikely to be improved to an acceptable level by purely physical factors such as particle size, porosity, etc.”

There are some minor omissions in the figures as well. It would be better to denote the peak positions in figs 2 b,d (XPS and Raman spectra). The same to the other figures as well.

In addition, there are several misprints/inappropriate wordings throughout:

Line 32: alternative to carbon materials anodes ---> alternative anodes

Line 44: delete the phrase: in terms of pore size

Line 141: different current intensities ---> different current densities

Line 147: tilted 71o respect to… ---> tilted at 71o with respect to

Line 226: to analyze in a detail ---> to analyze in detail

Line 232: oxidation-reduction process located at ---> redox reactions at the electrode surface

Line 234: Post-Morten ---> Post-Mortem

Line 337: vibrations bands ---> vibration bands

Line 354: capacity stability ---> cycle life

Author Response

We agree with the reviewer that the formation of relatively large Ni metallic islands is one of the main causes of the limited reversibility of these conversion systems. However, we consider that the nano-columnar morphology of the NiO sheets significantly favors the contact between the active material and the electrolyte, facilitating the movement of sodium ions and therefore the conversion reaction. On the other hand, if the porosity is adequate in terms of volume, pore size and interconnectivity, the accessibility to the active material will be favored by improving the electrochemical behavior not only in terms of capacity values but also of cycling rate. The last sentence of the conclusions has been modified to clarify this point as much as possible.

New annotations have been incorporated in some figures to identify the different peaks, and errors in the text have been corrected.

Round 2

Reviewer 2 Report

The authors have addressed all the issues, and the article can be accepted for publication in Nanomaterials.